# Human Tonsil-Derived Mesenchymal Stromal Cells Maintain Proliferating and ROS-Regulatory Properties via Stanniocalcin-1

**DOI:** 10.3390/cells9030636

**Published:** 2020-03-06

**Authors:** Yoojin Seo, Tae-Hoon Shin, Ji-Su Ahn, Su-Jeong Oh, Ye Young Shin, Ji Won Yang, Hee Young Park, Sung-Chan Shin, Hyun-Keun Kwon, Ji Min Kim, Eui-Suk Sung, Gi Cheol Park, Byung-Joo Lee, Hyung-Sik Kim

**Affiliations:** 1Department of Life Science in Dentistry, School of Dentistry, Pusan National University, Yangsan 50612, Korea; amaicat24@naver.com (Y.S.); anjs08@naver.com (J.-S.A.); dhtnwjd26@naver.com (S.-J.O.); bubu3935@naver.com (Y.Y.S.); midnightnyou@naver.com (J.W.Y.); 2Dental and Life Science Institute, Pusan National University, Yangsan 50612, Korea; 3Biomedical Research Institute, Pusan National University Hospital, Busan 49241, Korea; thshin1125@gmail.com (T.-H.S.); 24hera@hanmail.net (H.Y.P.); 4Translational Stem Cell Biology Branch, National Heart, Lung and Blood Institute, National Institutes of Health, Bethesda, MD 20892, USA; 5Department of Otorhinolaryngology, College of Medicine, Pusan National University and Biomedical Research Institute, Pusan National University Hospital, Busan 49241, Korea; cha-nwi@daum.net (S.-C.S.); kwon-h-g@hanmail.net (H.-K.K.); ny5thav@hanmail.net (J.M.K.); 6Department of Otorhinolaryngology-Head and Neck Surgery, Biomedical Research Institute, Pusan National University School of Medicine, Yangsan Pusan National University Hospital, Yangsan 50612, Korea; ch4oh@hanmail.net; 7Department of Otolaryngology – Head and Neck Surgery, Samsung Changwon Hospital, Sungkyunkwan University School of Medicine, Changwon 51353, Korea; uuhent@gmail.com

**Keywords:** tonsil mesenchymal stromal cells, stanniocalcin-1, proliferation, reactive oxygen species, inflammasome

## Abstract

Mesenchymal stromal cells (MSCs) from various sources exhibit different potential for stemness and therapeutic abilities. Recently, we reported a unique MSCs from human palatine tonsil (TMSCs) and their superior proliferation capacity compared to MSCs from other sources. However, unique characteristics of each MSC are not yet precisely elucidated. We investigated the role of stanniocalcin-1 (STC1), an anti-oxidative hormone, in the functions of TMSCs. We found that STC1 was highly expressed in TMSC compared with MSCs from bone marrow or adipose tissue. The proliferation, senescence and differentiation of TMSCs were assessed after the inhibition of STC1 expression. STC1 inhibition resulted in a significant decrease in the proliferation of TMSCs and did not affect the differentiation potential. To reveal the anti-oxidative ability of STC1 in TMSCs themselves or against other cell types, the generation of mitochondrial reactive oxygen species (ROS) in TMSC or ROS-mediated production of interleukin (IL)-1β from macrophage-like cells were detected. Interestingly, the basal level of ROS generation in TMSCs was significantly elevated after STC1 inhibition. Moreover, down-regulation of STC1 impaired the inhibitory effect of TMSCs on IL-1β production in macrophages. Taken together, these findings indicate that STC1 is highly expressed in TMSCs and plays a critical role in proliferating and ROS-regulatory abilities.

## 1. Introduction

Mesenchymal stem cells (MSCs) are multipotent stromal cells capable of differentiating into cells of mesenchymal lineages and can be isolated from most of post-natal organs and tissues including bone marrow, adipose tissue, liver, skin, umbilical cord and placenta [1,2,3]. Although various cell types can be differentiated from MSCs, osteoblast differentiation has been intensively studied to elucidate the pathogenesis of bone-related diseases and to develop MSC-based therapeutics. More recently, a number of preclinical and clinical studies including our own reports have demonstrated that MSCs can be utilized for therapeutic purpose against immune disorders because they possess immunomodulatory or anti-inflammatory abilities [4,5,6,7,8,9]. Despite the therapeutic application of MSCs for decades, the outcomes of clinical trials have been controversial and the diversity in cellular characteristics according to different tissue sources has been regarded as one of the major hurdles for the optimization and standardization of MSC therapy. Therefore, the potential distinctive characteristics among MSCs from different sources should be persistently studied to accumulate more references in this field, especially for recently reported novel sources [10,11,12,13].

Palatine tonsils have been introduced as novel promising tissue sources for MSCs, since it can be obtained easily by tonsillectomy and be utilized for further stem cell banking [14,15,16,17]. Several groups including our group have demonstrated common or unique characteristics of TMSCs and elucidated underlying mechanisms [15,16,17,18]. Moreover, a number of evidence have proven that TMSCs can be applied for therapeutic purposes [19,20,21]. In our previous study, we found that TMSCs reside in the perivascular area of palatine tonsil and they express W5C5, a novel marker for endometrial stem cells [18]. In addition, we reported that TMSCs express a much higher level of fibroblast growth factor (FGF)-5 when compared to adipose tissue-derived MSCs (AMSCs) or bone marrow-derived MSCs (BMSCs) and FGF5 might be one of the critical factors for the superior proliferative potential of TMSCs [17]. More recently, we verified the regenerative potential of TMSCs for the wound healing process, mostly exerted by their anti-inflammatory and anti-fibrotic functions [22]. Therefore, we further sought to explore key factors and regulatory mechanisms for beneficial properties of TMSCs based on our microarray analysis in which we compared the gene expression profile of TMSCs compared with AMSCs and BMSCs (GEO accession number: GSE77272).

Among the gene expression profile of TMSCs that we analyzed, STC1 expression was significantly higher in TMSCs than AMSCs or BMSCs. STC1 is a homologue of a hormone stanniocalcin that was first discovered in bony fishes as a regulatory glycoprotein for calcium/phosphate [23]. Although the mode of action by STC-1 has not been thoroughly identified, mammalian STC-1 has been reported to exert anti-oxidative effects through the reduction of reactive oxygen species (ROS) [24,25,26,27], leading to the attenuation of subsequent apoptosis [24,28]. A study by Ono et al. showed that mitochondria-related hormone STC1 was produced by MSCs in a paracrine fashion under stress conditions, which improved the cell survival through the upregulation of uncoupling protein 2 [29]. In addition, MSC-derived STC1 promoted cell survival by eliminating oxidative phosphorylation, reducing intracellular ROS, and converting metabolism into a more degraded metabolic profile [25,30]. Oh et al. uncovered that MSCs can inhibit the activation of nucleotide-binding domain and leucine-rich repeat pyrin 3 (NLRP3) inflammasome in macrophages by the suppression of mitochondrial ROS generation via STC1 release in response to soluble factors from activated macrophages [31]. Although few studies have demonstrated the function of MSC-derived STC1, most of these studies focused on the paracrine effects of secreted STC1 on neighboring cells rather than its autocrine function.

Therefore, in this study, we sought to investigate whether STC1 can affect the proliferation, senescence and differentiation of TMSCs, as well as the autocrine or paracrine regulation of oxidative stress in TMSCs or against macrophages, respectively. To verify the ROS-regulating function on macrophages, we observed the role of MSC-secreted STC1 against NLRP3 inflammasome activation in macrophage-like cells.

## 2. Materials and Methods

### 2.1. Isolation and Culture of MSCs

All the study protocols were approved by the Institutional Review Board of Pusan National University Hospital. Informed consent was obtained from all patients and their parents. The palatine tonsils were obtained from four patients with chronic tonsillitis. Tonsil tissues were washed with phosphate-buffered saline (PBS) followed by the digestion in 0.075% collagenase type I (Sigma-Aldrich, St. Louis, MO, USA) at 37 °C for 30 min. After the neutralization of enzyme activity with alpha-modified Eagle’s medium (α -MEM) containing 10% fetal bovine serum (FBS), the sample was centrifuged and the pellet was filtered through a 100-μm mesh. Cells were incubated overnight in control medium (a-MEM, 10% FBS, 100 U/mL penicillin, and 100 mg/mL streptomycin) and the attached cells were maintained after washing with PBS. To induce cellular senescence, etoposide (Sigma-Aldrich, St. Louis, MO, USA), one of FDA-proved chemotherapeutic agent, was treated at low concentration (100 nM and 200 nM) for 48 to 72 h. Replicative senescence of MSCs was also induced by repeated subculture every 3-4 days when the cells reach 80% confluence (by passage 24). For the present study, MSCs derived from tonsil tissues of two different donors were used.

AMSCs and BMSCs were isolated and characterized as described previously [32,33]. We obtained adipose tissues from four different patients who received the abdominoplasty. Bone marrow samples were obtained from four patients. Mononuclear cells were isolated from bone marrow samples by density-gradient centrifugation. Attached cells from digested adipose tissue or bone marrow mononuclear cells were maintained.

### 2.2. Microarray Analysis

To analyze the gene expression levels in TMSCs, AMSCs, and BMSCs, total RNA was isolated from each MSC from 4 different donors, respectively, using an RNeasy Mini Kit (QIAGEN, Valencia, CA, USA). Illumina HumanHT-12 BeadChip v4 (Illumina, Inc., San Diego, CA, USA) was used to conduct gene expression profiling. Genes related with anti-oxidative or –apoptotic functions were selected and heatmaps with hierarchical clustering were obtained using Multi Experiment Viewer (MeV) version 4.9.0 from the TM4 Microarray Software Suite (http://www.tm4.org/mev.html).

### 2.3. RNA Interference

Transfection of Small interfering RNA (siRNA) into TMSCs was conducted when the cells reached a confluency of 60~70%. siRNA oligonucleotide duplexes (ON-TARGETplus SMARTpool) for STC1 mRNAs, and non-targeting control (ON-TARGETplus siCONTROL) were purchased from Dharmacon (Thermo Scientific, Rockford, IL, USA). DharmaFECT1 (Thermo Scientific, Rockford, IL, USA) as a transfection reagent and siRNA at a concentration of 50 nmol/L were used. The transfection procedure was conducted according to the manufacturer’s instructions. Briefly, siRNA and transfection reagent-containing media (without antibiotics) were added when the cells reached the appropriate confluency. After 48 h, the medium was changed and incubated for an additional 24 h. Transfected cells were washed, detached and used for the following studies.

### 2.4. Cell Proliferation Assessment

To assess cell proliferation based on the detection of cell viability, MTT assay and CCK8 assay was conducted. For MTT assay, cells were incubated in fresh medium containing 200 μg/mL of MTT reagent (Amresco, Solon, OH, USA) for 4 h at 37 °C. After medium was removed, DMSO was added into each well and incubated with shaking for approximately 2~3 min. For CCK8 assay, CCK8 solution (Dojindo, Rockville, MD, USA) was added to the culture medium directly then incubated for 1 h at 37 °C. Finally, the absorbance at 450 nm was measured with a spectrophotometer.

### 2.5. Quantitative Real-time PCR (qRT-PCR)

For qRT-PCR, previously isolated RNA was reverse transcribed using M-MLV reverse transcriptase (Promega, Madison, WI, USA). cDNA was mixed with SYBR Green PCR Master Mix (Applied Biosystems, Foster City, CA) and primers followed by the amplification using an ABI 7500 real-time PCR system (Applied Biosystems, Foster City, CA, USA). Relative amounts of mRNA for each gene were determined by the comparative Ct method. The expression levels of target genes were normalized using the housekeeping gene, GAPDH. The expression levels of STC1, STC2, p16, p21, superoxide dismutase (SOD)-1, SOD2, peroxiredoxin (PRDX)-1, and glutathione peroxidase (GPX)-1 were calculated. Primer sequences were as follows in Table 1:

### 2.6. Apoptosis Assay

To determine the rate of apoptotic cells, Apoptosis Detection Kits (BD Bioscience, San Jose, CA, USA) were used. Cells were washed twice with PBS and resuspended in binding buffer. The mixtures of cells with annexin V and propidium iodide (PI) were gently vortexed and incubated for 15 min at room temperature in the dark. Then, binding buffer was added to the mixtures, and all samples were analyzed by flow cytometry, which was performed on a FACSCanto II (BD Bioscience, San Jose, CA, USA).

### 2.7. Live/Dead Staining

To evaluate the live/dead status of cultured cells, LIVE/DEAD™ Viability/Cytotoxicity Kit (Thermo Scientific, Rockford, IL, USA) was conducted as manufacturer’s instruction. Briefly, chemical treated- and control cells were incubated with 2 µM calcein AM (green fluorescence for the live cell labeling) and 4 µM EthD-1 (red fluorescence for the dead cell labeling) for 30 min at room temperature. Labeled cells were evaluated with fluorescence microscope.

### 2.8. β-galactosidase Staining

To detect the senescence of TMSCs, beta-galactosidase (β-gal) staining was performed as previously described [34]. Briefly, TMSCs were plated in 6 well plates and incubated till 70–80% confluency. After washing with PBS, the cells were fixed with glutaraldehyde (0.5%, pH 7.2) for 5 min at room temperature. Cell were washed with PBS containing MgCl2 followed by the staining with X-gal solution containing 1 mg/mL X-gal, 0.12 mM K3Fe(CN)6, 1 mM MgCl2 overnight at 37 °C. Stained images were captured after washing.

### 2.9. TMSC Differentiation

#### 2.9.1. Osteogenic Differentiation

For differentiation into osteoblasts, untreated or siRNA-transfected viable TMSCs were seeded in 6 well plates at 4 × 10^5^/well to reach 70–80% confluency. The cells were cultured with DMEM low glucose medium (10% FBS) containing 0.1 μM dexamethasone, 10 mM beta-glycerophosphate, and 50 μM ascorbate. The cells were maintained for 10, 15 and 20 days, with the replacement of media twice a week. Osteoblasts were detected by Alizarin Red staining. Photographs were taken and optical density was measured at 570 nm.

#### 2.9.2. Adipogenic Differentiation

For differentiation into adipocytes, untreated or siRNA-transfected viable TMSCs were seeded in 6 well plates at 4 × 10^5^/well to reach 70–80% confluency. The cells were cultured with DMEM low glucose medium (10% FBS) containing 1 M dexamethasone, 10 μg/mL insulin, 0.5 mM 3-isobutyl-1-methylxanthine, and 0.2 mM indomethacin. The cells were grown for 10, 15 and 20 days, with the replacement of media twice a week. Lipid droplet in adipocytes was detected by Oil Red O staining. Photographs were taken and optical density was measured at 500 nm.

### 2.10. ROS Measurement

MitoSOX (Thermo Scientific, Rockford, IL, USA) was used to measure mitochondrial superoxide. Untreated MSCs or tert-Butyl hydroperoxide (tBHP)-treated MSCs were incubated with MitoSOX at 37 °C for 30 min, then washed with PBS, and centrifuged. The cells were re-suspended in PBS and were analyzed by flow cytometry (BD Biosciences, Franklin Lakes, NJ, USA). Inhibitors for catalase (3-Amino-1,2,4-triazole) and silver diethyldithiocarbamate were purchased from Sigma Aldrich (St. Louis, MO, USA).

### 2.11. Induction of Macrophage-like Cells and NLRP3 inflammasome Activation

THP-1 cells (ATCC, Manassas, VA, USA), human acute monocytic leukemia cell line, were differentiated into macrophage-like cells. THP-1 cells were seeded in 6 well plates at 1 × 10^6^ cells/well in RPMI-1640 medium and treated with 200 nM phorbol 12-myristate 13-acetate (PMA, Sigma Aldrich, St. Louis, MO) for 48 h. Cells were washed with PBS, followed by the stabilization with fresh RPMI-1640 medium for additional 2 days. For NLRP3 inflammasome activation, differentiated macrophage-like cells were primed with 1 μg/mL of LPS for 4 h and, followed by stimulation with 5 mM of ATP (Invivogen, San Diego, CA, USA) for 45 min. TMSCs at 1 × 10^5^ cells/well (TMSC:THP-1 = 1:10) were added when Lipopolysaccharide (LPS) or Adenosine triphosphate (ATP) were treated to THP-1-derived macrophage-like cells. The production of IL-1β in culture supernatant after NLRP3 activation was measured by enzyme-linked immunosorbent assay (ELISA) using an enzyme linked immunosorbent assay kit (R&D Systems, Minneapolis, MN, USA).

### 2.12. Western Blotting Analysis

Cultured cells were washed in cold PBS and lysed in RIPA buffer (Sigma-Aldrich, St. Louis, MO, USA). After quantification, 15ug of proteins were loaded into a 10% SDS-polyacrylamide gel then transferred to nitrocellulose membranes (Amersham Pharmacia Biotech, Piscataway, NJ, USA), and probed with primary antibodies and detected using anti-mouse and anti-rabbit peroxidase-conjugated secondary antibodies (Thermo Scientific, Rockford, IL, USA) and visualized by enhanced chemiluminescence (Amersham Pharmacia Biotech, Piscataway, NJ, USA).

### 2.13. Statistical Analysis

The mean values of the different groups were expressed as the mean ± SD. All statistical comparisons were made using one or two-way ANOVA followed by the Bonferroni post-hoc test for multi-group comparisons using the GraphPad Prism version 5.01 (GraphPad Software, San Diego, CA, USA). Statistical significance designated as asterisks is indicated in the figure legends.

## 3. Results

### 3.1. TMSCs Highly Express STC1 and Cell Proliferation is Decreased after STC1 Inhibition

To investigate whether TMSCs possess unique anti-oxidative or –apoptotic functions, gene expression patterns between MSCs from different sources were analyzed using microarray. We observed that TMSCs demonstrated distinct gene expression patterns compared to AMSCs and BMSCs (Figure 1A). In particular, the expression of STC1 gene was highly expressed in TMSCs, 4 fold and 3.6 fold higher compared to both AMSCs and BMSCs, respectively (Figure 1A). Among well-known ROS-regulatory factors expressed in MSCs, the expression levels of SOD1, SOD2, and PRDX1 in TMSCs were similar to those in AMSCs. Although GPX1 level was determined to be higher in TMSCs, the fold increase was less than 2 fold. On the other hand, mRNA levels of STC1 and STC2 in TMSCs were elevated to 31 and 2.3 fold, respectively (Figure 1B). These differences in STC1 and STC2 expression were confirmed by immunoblotting, however, basal expression of STC2 on protein level was not detectable (Figure 1C). Therefore, we next examined the role of STC1 in TMSC functions. The expression of STC1 was down-regulated by siRNA transfection and determined by qPCR (Figure 1D). Interestingly, STC1 inhibition significantly decreased the proliferation of TMSCs (Figure 1E,F). Given that the expressions of p16INK4A and p21CIP1/WAF1, cyclin-dependent kinase inhibitors, are strongly affected by the ageing of cancer cells or MSCs [35,36,37], we next determined the expression of these genes in TMSCs after STC1 inhibition. The expressions of p16 and p21 in mRNA level were significantly up-regulated in TMSCs treated with siRNA for STC1 (Figure 2A). The increase in these senescence-related factors was confirmed by immunoblotting (Figure 2B). Next, we performed β-gal staining and confirmed that STC1 inhibition led to cellular senescence (Figure 2C). Importantly, the proportion of apoptotic cells in TMSCs was not altered by the suppression of STC1 expression (Figure 2D,E). Taken together, these results indicate that STC1 plays a critical role in the regulation of proliferation and ageing of TMSCs without the induction of apoptosis.

### 3.2. STC1 Expression is not Altered in Chemically Induced Senescent TMSCs

We next investigated whether the expression level of STC1 is associated with TMSC ageing process. To induce the senescence in TMSCs, etoposide was treated to TMSCs at low concentration as reported previously [38]. Upon etoposide treatment, cell proliferation rate assessed by CCK8 assay was decreased in a concentration-dependent manner (Figure 3A), while cell viability was not altered (Figure 3B). TMSCs cultured with etoposide exhibited enlarged cell body with a flattened shape, as well as increased staining for β-gal (Figure 3C). Molecular analysis of p16 and p21 expression also confirmed that etoposide could lead to TMSC senescence in vitro (Figure 3D). To determine the causal relationship between senescence and STC1 expression in TMSCs, we detected STC1 expression in TMSCs in the presence of etoposide; however, STC1 protein level was not altered by etoposide treatment (Figure 3E). In addition, we also induced replicative senescence of TMSCs and confirmed that cell proliferative capacity was decreased over repeated subculture (Figure 3F–G) while the proportion of β-gal positive senescent cells was significantly increased (Figure 3H). In line with etoposide treated cells, however, STC1 protein level was not changed after a series of passaging from p2 to p24 (Figure 3I). These findings imply that STC1 inhibition might induce the ageing of TMSCs, but STC1 expression would not be affected by cellular senescence in vitro.

### 3.3. STC1 is not Involved in Differentiation Potential of TMSCs

To determine whether STC1 is involved in osteogenic or adipogenic differentiation of TMSCs, the cells were treated with siRNA for STC1 and differentiation was induced using conditioned medium for each differentiation. During osteogenic or adipogenic differentiation of TMSCs, STC1 did not affect the intensity of the Alizarin Red S or Oil Red O staining, respectively (Figure 4A–D). These results suggest that STC1 is not involved in the differentiation potential of TMSCs into osteoblasts or adipocytes.

### 3.4. STC1 is Pivotal for the Maintenance of ROS Homeostasis, as Well as the Regulation of tBHP-Induced ROS Production in TMSCs

STC1 secreted from the MSCs has been reported to promote cell survival by reducing ROS. Therefore, we next examined STC1-mediated regulatory functions of TMSCs on ROS generation in TMSC themselves compared to reported ROS-regulatory enzymes expressed in MSCs from various sources including catalase and superoxide dismutase (SOD). The proportion of MitoSOX-positive cells was measured to determine cells containing increased mitochondrial ROS. The basal level of ROS was not altered when inhibitors of catalase or SOD were treated. However, STC1 inhibition resulted in a significant elevation of ROS-positive cells to 6.7% (Figure 5A,B). Moreover, the combination of STC1 and catalase inhibition increased ROS level to 10.4% (Figure 5A,B). To more elucidate the role of STC1 in the environment where oxidative stress is present, TMSCs were treated with tBHP and ROS level was measured in the presence of each inhibitor or siRNA. ROS generation in TMSCs was elevated to approximately 5.65% when tBHP was treated. Interestingly, STC1 inhibition remarkably increased ROS level to 18.7%, whereas inhibitors for catalase or SOD did not affect the level (Figure 5C,D). Our data indicate that STC1 is critically involved in the regulation of both basal and tBHP-induced ROS generation in TMSC.

### 3.5. TMSCs Can Suppress ROS-Mediated Activation of NLRP3 Inflammasome in Macrophages

Given that MSC can inhibit ROS-mediated activation of NLRP3 inflammasome, we investigated whether TMSCs can exhibit a similar effect. Human THP1-derived macrophages were primed with LPS for 4 h and then stimulated with ATP for 45 min to activate the NLRP3 inflammasome. TMSCs were co-cultured with THP1-derived macrophages beginning from the priming or stimulation. IL-1β secretion was elevated in inflammasome-activated macrophages and was significantly reduced by co-culture with TMSCs (Figure 6A,B). Because IL-1β secretion was more consistently suppressed when TMSCs were added at the point of LPS treatment (Figure 6A,B), co-culture of TMSC-macrophage beginning from LPS priming was performed to verify the influence of STC1 on this inhibitory effect of TMSCs against NLRP3 inflammasome activation. Interestingly, IL-1β secretion was restored when TMSCs were treated with siSTC1 (Figure 6C). To investigate whether this inhibitory effect of STC1-mediated TMSCs on IL-1β production in macrophages, we determined ROS level in macrophages after inflammasome activation in the presence or absence of TMSCs. NLRP3 inflammasome activation led to the increased level of ROS and elevated ROS level was further restored when TMSCs were co-cultured. Interestingly, STC1 inhibition partially suppressed this ROS-regulatory property of TMSCs (Figure 6D). Our findings suggest that TMSC-secreted STC1 can suppress NLRP3 inflammasome activation in macrophages by suppressing ROS generation.

## 4. Discussion

ROS is one of the crucial signaling molecules for the regulation of cell metabolism, proliferation and survival [39]. An increased generation of ROS more than basal level leads to alterations of signaling proteins and subsequent functional abnormalities. Wang et al. reported that several signaling proteins in stem cells are regulated by ROS, resulting in the alteration in pluripotency, self-renewal, survival and even genomic stability [40]. More importantly, MSCs are known to possess antioxidant properties [41]. MSCs are reported to have relatively low levels of intracellular ROS and high levels of anti-oxidative enzymes, such as SOD1, SOD2, and GPX1 [41,42]. Therefore, in the present study, we analyzed expression profiles of genes for ROS-related enzymes or signaling among MSCs from three different sources. We found that STC1 was highly expressed in TMSCs. STC1 inhibition resulted in a significant decrease in TMSC proliferation. One might envision that elevated level of basal intracellular ROS might suppress cell proliferation. In our results showing ROS level in TMSCs after inhibition of several candidate factors with anti-ROS function, STC1 inhibition significantly increased the basal level of mitochondrial ROS. Therefore, STC1-mediated ROS regulation might be involved in the proliferation of TMSCs. Moreover, previous studies suggested that STC1 can regulate cell proliferation via a variety of critical cellular signaling pathways. A study by Bai et al. demonstrated that STC1 promotes cell proliferation in human prostate carcinoma via cyclin E1/cyclin-dependent kinase 2 [43]. On the contrary, STC1 was reported to inhibit cell proliferation in cervical cancer cells through NF-κB p65 activation [44]. These conflicting results suggest that further studies to investigate the role of STC1 in each cell type are required. Since several groups previously revealed that cyclin-dependent kinase (CDK) inhibitors including p16INK4A, p19Arf, and p21CIP1/WAF1 are involved in the proliferation and ageing of MSCs [37,45], we determined the change in the expression of these CDK inhibitors and observed the consistent results correlating with the decrease in cell proliferation. Although STC1 inhibition induced the increase in the expressions of p16INK4A and p21CIP1/WAF1, which might indicate the cellular senescence, STC1 expression was not altered in chemical-mediated senescent cells. Because precise mechanisms of ageing among old donor-derived cells, chemical-induced senescent cells, and long-term cultured, late-passage cells are quite different, a further study using MSCs from old donors may be required to elucidate the discrepancy in our findings.

A number of studies have proposed a relationship between ROS and osteogenic differentiation. Most of these studies suggest that ROS inhibits the osteogenesis [46,47,48]. However, some studies report that ROS promotes calcification and osteogenesis, suggesting that at least a basal level of ROS might be required for regulation of MSC fate [49]. Regarding adipogenic differentiation, ROS reduces adipogenesis and its level is elevated as MSCs differentiate into adipocytes [50,51,52]. In the present study, STC1 inhibition did not affect osteogenic or adipogenic differentiation of TMSCs. Considering that ROS level in TMSCs was elevated when STC1 was inhibited, ROS level altered by STC1 inhibition might have a marginal influence on the differentiation of TMSCs.

In the present study, STC1 was not only involved in the regulation of homeostatic or oxidative stress-mediated ROS generation, but also in the suppression of inflammasome-mediated production of ROS in macrophage-like cells. Inflammasomes are multimeric protein complexes, mainly present in antigen-presenting cells which contribute to innate immunity. When activated, these complexes activate caspase-1 to produce active IL-1β [53,54]. Among several types of inflammasomes, NLRP3 inflammasome has been most frequently studied since this inflammasome has a unique activation process and is involved in various intractable diseases. Mitochondrial ROS has been reported to be one of the important upstream signals for NLRP3 inflammasome activation and a number of studies have demonstrated that ROS scavengers can suppress NLRP3 inflammasome activation [55,56]. In a previous study by Prockop group, STC1 produced from BMSCs remarkably inhibited the IL-1β secretion by NLRP3 activation via down-regulation of ROS generation [31]. Consistently in the present study, TMSCs also significantly suppressed IL-1β secretion from THP1-derived macrophage-like cells following the treatment of LPS and ATP for NLRP3 inflammasome activation. And STC1 inhibition completely rescued this inhibitory effect of TMSCs on IL-1β production. These findings along with the results of ROS measurement in TMSC themselves indicate that STC1 might be critically involved in the ROS regulation in macrophages but redundant in TMSCs. A further study revealing the expression patterns of ROS-regulating enzymes in MSCs compared to macrophages might be required to precisely elucidate this discrepancy.

Although MSC therapy has its limitations in that they exert different therapeutic potency depending on sources or donors, only a few studies have tried to uncover the underlying mechanisms for this variation in their potency. And most of the previous studies only listed the results of mRNA or miRNA analysis and did not point out key factors and signals nor experimentally prove their functional outcomes. In our study, we confirmed higher expression of STC1 in TMSCs compare to AMSCs, along with representative ROS-related factors in MSCs and demonstrated that STC1 is critical for ROS homeostasis of TMSCs. This abundant expression of potent anti-oxidative hormone in TMSCs might highlight these MSCs from relatively novel source, a palatine tonsil, as a promising therapeutics which can engraft more stably and survive longer after administration into the lesion.

Taken together, our findings in the present study summarize that TMSCs uniquely express STC1 at a higher level than MSCs from other sources and STC1 critically contributes to the maintenance of proliferation of TMSCs, as well as their ROS-regulatory and ROS-resistant properties.

## Figures and Tables

**Figure 1 cells-09-00636-f001:**
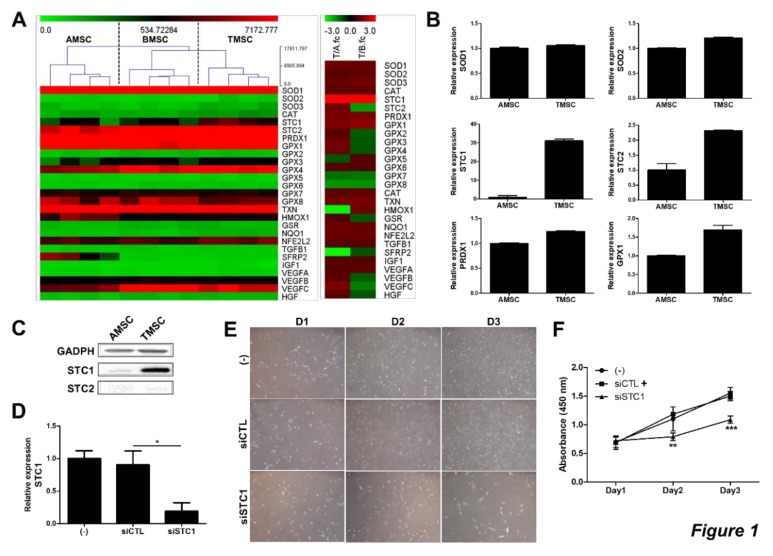
Functional expression of STC1 in TMSCs). Gene expression profiles for ROS-regulating factors were analyzed in TMSCs compared to BMSCs and AMSCs. T/A fc and T/B fc in right panel indicate fold change of TMSC/AMSC and that of TMSC/BMSC, respectively (**A**). mRNA expressions of pivotal factors for ROS regulation in TMSCs were determined by qPCR compared with those in AMSCs (**B**). STC1 and 2 expressions in protein level were detected by immunoblotting (**C**). siRNA for STC1 was transfected in TMSCs and cell viability was measured by MTT assay (**D**). Photographs of cells were taken at day 1, 2 and 3 after seeding of untreated TMSCs and siCTL- or siSTC1-treated TMSCs (**E**). Cell viability of siSTC1-treated TMSCs was determined by MTT assay compared to untreated or siCTL-treated TMSCs (**F**). Results are three technical replicates of TMSC from one donor. Representative results from two different TMSCs with similar tendency were presented. **P*< 0.05, ***P* < 0.01, ****P* < 0.001. Results are shown as mean ± SD.

**Figure 2 cells-09-00636-f002:**
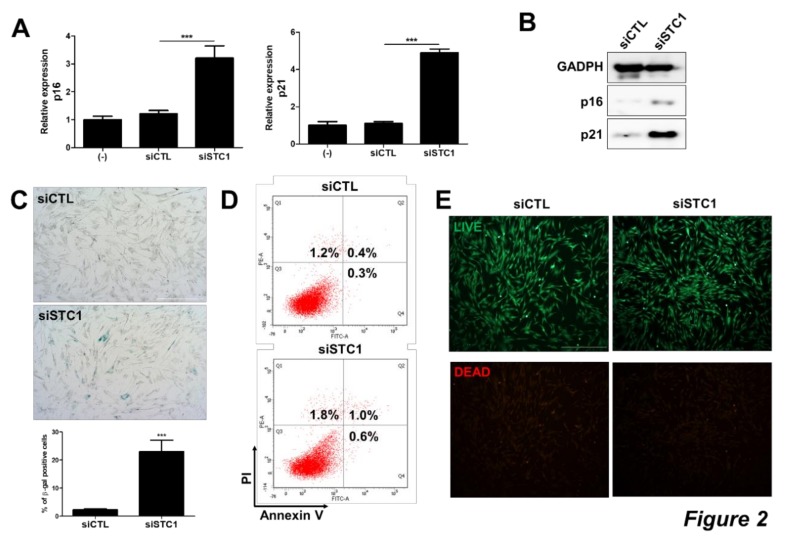
Induction of TMSC senescence by STC1 inhibition. After three days of siSTC transfection, the expressions of cyclin dependent kinase inhibitors in TMSCs were determined in mRNA level by qPCR (**A**) and protein level by immunoblotting (**B**). Cellular senescence was assessed by β-gal staining and the number of β-gal positive cells compared to control group was counted (**C**). Annexin V and PI were stained in untreated or siSTC-treated TMSCs and analyzed for apoptosis by flow cytometry (**D**). Cell viability was evaluated by Live/Dead staining (**E**). Results are three technical replicates of TMSC from one donor. Representative results from two different TMSCs with similar tendency were presented. ****P* < 0.001. Scale bar = 500 μm. Results are shown as mean ± SD.

**Figure 3 cells-09-00636-f003:**
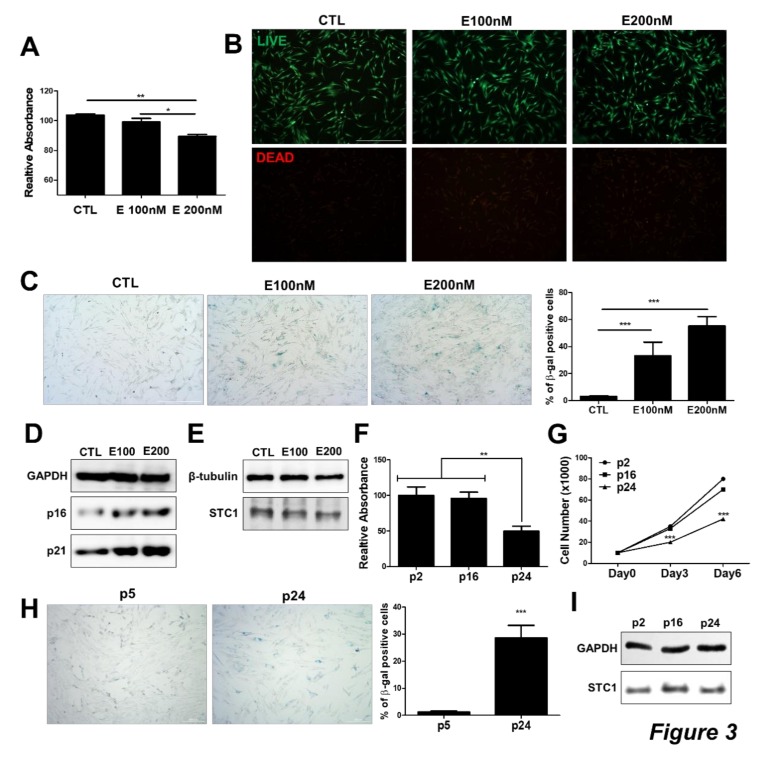
STC1 expression in etoposide-mediated senescent TMSCs. To induce senescence in TMSCs, etoposide was treated to TMSCs for 3 days then cellular senescence, as well as viability, was analyzed. Cell viability was measured by CCK8 assay (**A**) and Live/Dead assay kit (**B**).Cellular senescence was determined by staining for β-galactosidase (**C**). protein levels of cellular senescence markers and STC1 in TMSCs were detected by immunoblotting upon etoposide treatment (**D**,**E**). Replicative senescence was induced and cell viability and proliferative capacity was analyzed by MTT assay (**F**) and cell counting (**G**), respectively. The distribution of senescent cells was determined by β-galactosidase staining (**H**). STC1 protein levels at passage 2, 16, and 24 were assessed by immunoblotting (**I**). Results are three technical replicates of TMSC from one donor. Representative results from two different TMSCs with similar tendency were presented. **P* < 0.05, ***P* < 0.01, ****P* < 0.001. Scale bar = 500 μm) and 200 μm (H). Results are shown as mean ± SD.

**Figure 4 cells-09-00636-f004:**
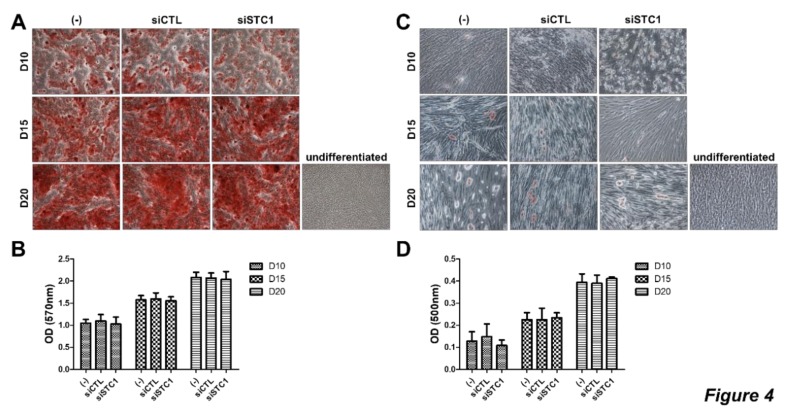
Differentiation of TMSCs after STC1 inhibition. Differentiation of TMSCs was evaluated by specific staining after STC1 inhibition (**A**–**D**). Osteogenic differentiation was induced in untreated or siCTL- and siSTC1-treated TMSCs and stained by Alizarin red S at day 10, 15, and 20, followed by the elution for quantification (**A**,**B**). Adipogenic differentiation was induces in TMSCs and stained with Oil Red O at day 10, 15, and 20, followed by the elution for quantification (**C**,**D**). Results are three technical replicates of TMSC from one donor. Representative results from two different TMSCs with similar tendency were presented. Results are shown as mean ± SD.

**Figure 5 cells-09-00636-f005:**
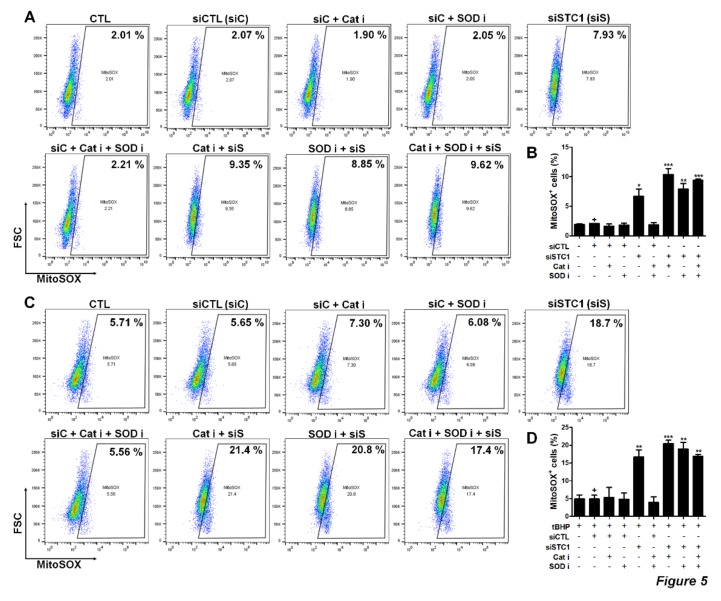
ROS regulation by STC1 in TMSCs. Mitochondrial ROS level in TMSCs after STC1, catalase or superoxide dismutase (SOD) inhibition was measured by staining with mitoSOX. mitoSOX+ cells were determined by flow cytometry (**A**) and quantified (**B**). Tert-Butyl hydroperoxide (tBHP) was treated for the induction of ROS generation in TMSCs and mitoSOX+ cells were determined by flow cytometry (**C**) and quantified (**D**). Dot plot data are representative results from two different TMSCs with similar tendency. Quantified data are integrated results of two technical replicates of TMSCs from two donors. **P* < 0.05, ** *P* < 0.005, *** *P* < 0.001. Results are shown as mean ± SD.

**Figure 6 cells-09-00636-f006:**
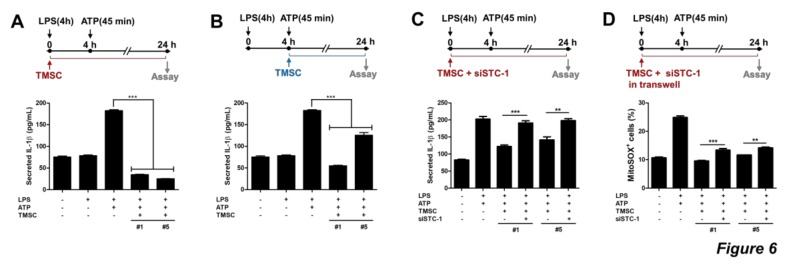
ROS regulation in THP1-derived macrophage-like cells by STC1. THP1-derived macrophage like cells were stimulated with LPS and ATP for NLRP3 activation and co-cultured with human TMSCs. IL-1β production was detected by ELISA (**A**–**C**). TMSCs were added to macrophages at LPS priming step (**A**) or at ATP stimulation step (**B**), and NLRP3 activation was assessed by IL-1β secretion. Untreated or siSTC1-treated TMSCs were co-cultured with THP1-derived macrophages from LPS priming step (**C**). After 24 h of co-culture, IL-1β secretion in the supernatants was quantified by ELISA (**A**–**C**). After 24 h of co-culture using transwell, mitochondrial ROS level was measured by staining with mitoSOX. mitoSOX+ cells were determined by flow cytometry and quantified (**D**) Results are three technical replicates of TMSCs from one donor. Representative results from two different TMSCs with similar tendency were presented. ***P* < 0.005, *** *P* < 0.001. Results are shown as mean ± SD.

**Table 1 cells-09-00636-t001:** Primer sequences for qRT-PCR.

Target Name	Forward Primer	Reverse Primer
STC1	GCAGGAAGAGTGCTACAGCAAG	CATTCCAGCAGGCTTCGGACAA
STC2	GCATGACTTTTCTGCACAACGCT	GGCTTATGCAGCCGAACCTGTG
SOD1	CTCACTCTCAGGAGACCATTGC	CCACAAGCCAAACGACTTCCAG
SOD2	CTGGACAAACCTCAGCCCTAAC	AACCTGAGCCTTGGACACCAAC
PRDX1	CTGCCAAGTGATTGGTGCTTCTG	AATGGTGCGCTTCGGGTCTGAT
GPX1	GTGCTCGGCTTCCCGTGCAAC	CTCGAAGAGCATGAAGTTGGGC
P16	CTCGTGCTGATGCTACTGAGGA	GGTCGGCGCAGTTGGGCTCC
P21	AGGTGGACCTGGAGACTCTCAG	TCCTCTTGGAGAAGATCAGCCG
GAPDH	GTCTCCTCTGACTTCAACAGCG	ACCACCCTGTTGCTGTAGCCAA

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
