# Peer review of "Human Tonsil-Derived Mesenchymal Stromal Cells Maintain Proliferating and ROS-Regulatory Properties via Stanniocalcin-1"

_cells, 2020, doi:10.3390/cells9030636_

Round 1

Reviewer 1 Report

The paper describes tonsil-derived mesenchymal stromal cell expression of  STC1 and involvement of this molecule in regulation of ROS. 

I strongly recommend to the Authors to change the description of isolated cells as "stem cells". Mesenchymal stromal cells is more appropriate term for these cells. In addition to this, the entire content of the manuscript doesn't provide any evidence of stemness of these cells. Authors only explored effects of STC1 knockdown effect in proliferation in vitro - this should not be commented as self-renewal as suggested by the title and lines 213 and 378. Also, please do not use term naive TMSC for cells subcultured in vitro (line 221).

General comments: Authors should indicate how many times are experiments repeated since statistical analysis looks unconvincing (Figure 5, 6). 

Figure 2. Can Authors provide any quantification of beta-gal activity?

Figure 3. One of the most important concerns regarding study design: If siRNA stronlgy affect proliferation rate of cells, how differentiation assays were performed? Also, control groups ( cells cultivated in growth medium) need to be shown. Oil Red staining is not convincing these photos. 

Figure 6 and cultivation of TMSC with THP1 cell line. Authors should explain in detail how this co-culture was assessed and  in which ratio cells were seeded. Moreover, Il-1beta level in graph A and C secreted by THP-1 cells (in presence of LPS and ATP only) is quite different. How Authors can explain this?

Discussion. Line 325. Authors here stated that STC1 inhibition did not significantly increase the basal level of mitoROS. This is conflicting with their statement in the Abstract (line 37-38) and Results (271-272). 

Author Response

We really appreciate the constructive and helpful comments of reviewers. We have carefully addressed the comments and have revised the manuscript as suggested. Our responses are given point-by-point manner below, in blue font. We made every effort to improve our manuscript based on the suggested comments.

Our point-by-point responses are attached as a pdf file.

Reviewer 2 Report

The authors characterised the in vitro properties of human tonsil-derived mesenchymal stem cells compared with bone marrow and adipose tissue-derived mesenchymal stem cells. They investigated the effect of STC1 on cell proliferation, survival, differentiation and cell homeostasis.

I found the topic well introduced and I really enjoyed reading it. The experiments were designed to simply address the authors hypothesis. 

Below minor comments:

1- I suggest to include in the methods, the starting number of cells used for the differentiation. This would help people interested in replicating results or just performing similar experiments. 

2- I suggest to include the list of the primers used.

3- I consider important to mention the number of biological replicates in the figure legends. 

4- There is an error in page8, line 271. The Reference to Fig. 6A and B is incorrect, it should be Fig. 5A and B. 

5- The resolution of the Fig. 1A is very bad. I suggest to improve it if possible.

Author Response

(The authors gave the same response as above.)

Reviewer 3 Report

The Authors investigated the role of  stanniocalcin-1 (STC1) on the proliferation, senescence and differentiation of TMSCs, and  the autocrine or paracrine regulation of oxidative stress in TMSCs and against macrophages.

Introduction: In the Introduction section the Authors state that  “Therefore, the potential distinctive characteristics among MSCs from different sources should be persistently studied…………..”.  Similar and distinctive characteristics of MSCs from different tissue sources are widely studied by different research groups and proper references should be added (eg. World J Stem Cells 11 (2019) 347-374, PMID 31293717; and/or Stem Cells Dev 2018. 27 (2), 65-84, and/or PMID 29267140; Cell Commun Signal 2011. 9:12 PMID 21569606).

Major comments:

Methods: The procedure of transfection of siRNA into TMSCs should be more detailed described.

Results: Analysis of ROS-regulatory factors expressed in MSCs could be better introduced in the results section, not only by description that “……….STC1 expression was approximately 30 fold higher in TMSCs…” or “……p16 and p21 in mRNA level were significantly up-regulated in TMSCs treated with siRNA for STC1” but exact value of relative gene expression, if significant, should be added (similarly to other examined gene expression if significant).

Figure 1 should be better prepared. On the Fig. 1A the gene expression profiles for ROS-regulating  factors are not clear which column illustrate TMSCs or BMSCs or AMSCs

The senescence in TMSCs was induced by etoposide treatment. It is well known that MSCs may lost biological activity with the number of passages and evaluation of STC1 expression in TMSCs after subsequent passages would be more relevant to their biological function than chemically induced senescence.

The assessment “…………..increased ROS level to a slightly greater extent (Fig.6A and B)” (line 273) is superficial and term “slightly extend” should be replaced by more appropriate wording and/or exact value of measured parameters. Moreover, this is illustrated on Fig 5A,B not on Fig 6AB.

Similarly, the term  “slightly elevated” should be corrected by proper wording in the Results and Discussion section.

Author Response

(The authors gave the same response as above.)

Round 2

Reviewer 1 Report

The revised Manuscript is really improved and can be considered for publication. Authors addressed my concerns. 

Reviewer 3 Report

Accept in present form